# The Influence of the Technological Process on Improving the Acceptability of Bread Enriched with Pea Protein, Hemp and Sea Buckthorn Press Cake

**DOI:** 10.3390/foods11223667

**Published:** 2022-11-16

**Authors:** Gabriela Daniela Stamatie, Iulia Elena Susman, Sabina Andreea Bobea, Elena Matei, Denisa Eglantina Duta, Florentina Israel-Roming

**Affiliations:** 1National Institute of Research & Development for Food Bioresources, IBA Bucharest, 021102 Bucharest, Romania; 2Faculty of Biotechnology, University of Agronomic Sciences and Veterinary Medicine of Bucharest, 59 Marasti Blvd, District 1, 011464 Bucharest, Romania; 3National Institute of Materials Physics, Atomistilor 405A, 077125 Magurele, Romania

**Keywords:** pea protein, hemp press cake, sea buckthorn press cake, protein enriched baked products, sensory analysis

## Abstract

The current consumption trends of plant based functional products have encouraged researchers and industry to study the production of protein enriched bakery products as a source of protein. In the context of the circular economy, the press cakes remaining after extraction of juices/oil from plants such as sea buckthorn or hemp can be valorized as they are rich in proteins, fibers and many bioactive compounds. Their use in bread making is a good solution to enrich the nutritional value of bread. Pea protein concentrate, hemp and sea buckthorn ingredients from press cakes by-products were added to whole wheat flour in different percentages and combinations (2% pea protein concentrate; 1% pea + 2% sea buckthorn ingredients; 1% pea + 2% hemp ingredients). Bread samples were obtained through three technological methods: one phase baking process (dough), two phases (sponge and dough) and one phase with dried sourdough added directly into the dough. A control sample (100% wheat whole flour) was considered. The mixtures of whole wheat flour and plant protein ingredients were rheologically tested. The bread samples were physicochemically analyzed (protein, fat, carbohydrates, energy value) and sensory characteristics were evaluated (texture, color and overall acceptability). The changes in the physicochemical characteristics, rheology behavior, microstructure and sensory quality were evaluated and compared. The energy from protein varied from 17.26 to 19.34% which means that all the samples can be considered “a source of protein”. Hardness decreased in samples with sponge and dried sourdough which reflect the importance of technology in keeping the freshness of the product. The most appreciated were the samples with pea protein concentrate, with hemp ingredient obtained through an indirect bread making process and the sample with sea buckthorn ingredient prepared through a direct bread making process using dried sourdough.

## 1. Introduction

There is a general global trend to reduce the use of animal protein and increase the consumption of plant proteins to protect the environment and reduce greenhouse gas emissions. Consumers have become more aware of what they eat and want nutritionally improved products, such as bread, with a higher protein content, which can supply nitrogen and amino acids to the body. Replacing some of the wheat flour in bakery products with different ingredients rich in proteins, fibers and polyphenols, especially in bread which is a fundamental food product, can deliver heathy compounds to the human body but this determines changes in the rheology and technological behavior of the dough during bread making [1].

In recent years, there has beenan increased demand for hemp protein foods as functional products, due to their recognized nutritional value, excellent digestibility and low allergenicity. Bread supplemented with hemp flour (>10%) guarantees a higher nutritional value, without negative effects on the dough stability or strength [2]. Moreover, hemp protein is superior to soy protein as a functional ingredient in the production of bread, because the dough formed with hemp protein has properties closer to that of wheat dough and the obtained bread has a good volume [2]. Hemp seed press cakes, a by-product from oil extraction, are rich in proteins and fiber and can be valorized as a protein rich ingredient in bread [3].

Besides hemp seed press cakes, sea buckthorn press cakes remaining after juice and oil extraction are still rich in vitamins, proteins and fibers, and can be valorized in breads or other bakery products bringing a specific and original taste. Studies were conducted on the use of *Hippophaë* species as medicine and food [4]. In the food industry, sea buckthorn is used raw, for obtaining functional foods or food supplements. The protein content varies between different parts of the sea buckthorn plant (woody green parts, seeds, leaves, bark, branches) [5,6]. The total protein content found in different sea buckthorn species was 46–129 g/kg dry weight (Indian variety) [7] and 93 g/kg dry weight (Polish variety) [8]. Sea buckthorn fruits are rich in essential aminoacids important for the human body [6,9,10,11,12,13].

Due to the positive effect on dough stability and water absorption, pea protein can be used to improve the protein content of baked products [14] at a substitution rate up to 10% of wheat flour. Pea protein has a low fat content and a low content of anti-nutritional compounds; it positively influences glycemic response, insulin resistance, cardiovascular health and gastrointestinal problems [15].

The purpose of the study was to evaluate three technological methods to valorise hemp and sea buckthorn press cakes (rich in proteins and fibers) for increasing the protein content of bread (with at least 12% of the energy value provided by protein in order to be a “source of protein”). Various tests were done to establish the optimum amount of the hemp and sea buckthorn ingredients which can be used in bread [15,16] as the addition of high amounts in wheat flour dramatically influences the quality of the dough and the overall sensorial attributes of the end-products. Because of the negative effects on sensory attributes, very small amounts of hemp and sea buckthorn protein ingredients (2–5%) could be added into the bread dough which did not sufficiently increase the protein content of the bread. For this reason, a very good quality plant protein concentrate from pea was selected to be combined with hemp and sea buckthorn ingredients from press cakes to balance the influence on the overall bread quality. Whole wheat flour was used as a basis for experiments due to the healthy compounds available (fibers, vitamins, minerals) but also due to the color and particle size which are more appropriate to allow the inclusion of hemp and sea buckthorn protein ingredients from press cakes which have a dark color and a coarse texture.

## 2. Materials and Methods

### 2.1. Materials

The following protein ingredients were used in the experiments: (1) pea protein concentrate (moisture 7.43%, protein 77.96%, fat 0.29%, ash 3.77%; distributed by Bio Holistic SRL, country of origin: Canada, processed and packaged in Great Britain, ecologically certified distributor from Oradea) [17]; hemp press cake powder as protein ingredient (moisture 4.93%, protein 29.65%, fat 11.88%, ash 5.77%, 13.29% total carbohydrates—Natural Ingredients R & D SRL, Făgăraș, Romania), obtained after the extraction of oil from hemp seeds [17]; (3) sea buckthorn press cake as protein ingredient (moisture 7.68%, protein 14.4%, fat 10.19%, ash 4.48%, 18.03% total carbohydrates—Natural Ingredients R & D SRL, Făgăraș, Romania), obtained after the extraction of oil/juice from sea buckthorn fruits [17]; whole wheat flour (moisture 11.03%, protein 14.92%, fat 1.9%, ash 1.36%, from the market); yeast and salt (commercially available). A dehydrated sourdough (containing a mixed of yeasts and lactic bacteria) was provided by Millbo SPA (Via Bellaria, Italy): 7–9% moisture, pH 4–5. 

### 2.2. Methods

#### 2.2.1. Dough Testing 

The rheological properties of whole wheat flour and the blends (whole wheat flour and protein ingredients) were studied using a Mixolab analyzer (Chopin Technologies, Villeneuve-la-Garenne, France) through AACC Method 54–60.01 [18]. The following samples were analyzed: C-control from whole wheat flour, WP-whole wheat flour + 2% pea concentrate, WPB-whole wheat flour + 1% pea concentrate + 2% sea buckthorn ingredient, WPH-whole wheat flour + 1% pea concentrate + 2% hemp ingredient. The dried mixtures were loaded into the Mixolab bowl and mixed with distilled water to produce 75 g dough, and the target consistency (torque) was 1.1 0.05 Nm. The dough was subjected to dual-mixing (80 rpm) during a heating and cooling program following the “Chopin+” protocol, namely: 8 min mixing at 30 °C, 4 °C/min heating to 90 °C, 7 min holding at 90 °C, 4 °C/min cooling to 50 °C, and 5 min holding at 50 °C. The Mixolab software (version 4.0.8) was used to record the curves and calculate the dough mixing parameters. The samples were analyzed at adapted hydration (i.e., the initial torque C1 was kept constant at 1.1 Nm). The parameters from the Mixolab curves refer to the following: dough development time, the time needed to attain a torque of 1.1 Nm (min); stability, the dough mixing resistance (min); C2, the torque associated with protein weakening (Nm); C3, the degree of gelatinization of the starch Appl. Ski. 2021, 11, 436 3 of 13 (Nm); C4, the stability of the starch gel formed (Nm); and C5, the retrogradation of the starch (Nm) [19]. The analysis was performed in duplicate.

#### 2.2.2. Bread Preparation

The formulations of the bread samples were based on whole wheat flour, which was replaced with different concentrations of protein-rich ingredients from peas, hemp and sea buckthorn (Table 1). The samples were: C-control from whole wheat flour; WP-whole wheat flour + 2% pea concentrate; WPB-whole wheat flour + 1% pea concentrate + 2% sea buckthorn ingredient; WPH-whole wheat flour + 1% pea concentrate + 2% hemp ingredient.

Bread samples were obtained through three technological methods:
(1)One phase/direct bread making process (dough);(2)Indirect bread making in two phases (sponge and dough);(3)Direct bread making process with dried sourdough added directly into the flour during the mixing phase.

The bread samples obtained using the ingredients and quantities from Table 1 are presented in Figure 1.

In the one phase/direct baking process, all the powdered ingredients were mixed, the yeast (3%) was solubilized in water and the salt (1.5%) was dissolved in water and added to the floury mixture. The amount of water calculated through Mixolab measurements (and adjusted if necessary) was added to the ingredients (660–690 mL water) to form through kneading an homogenous dough. The samples were coded as: C, WP, WPB, WPH.

In the indirect two phase process, a sponge was obtained using half of the dried ingredients (whole wheat flour and protein ingredients), yeast (2%) and water (1:1 floury mixture: water). The sponge was fermented for 90 min at 30–35 °C, covered with a plastic foil. Then, the remaining half of the flour, the salt (1.5%) dissolved in water and the rest of the water were mixed in to obtain an homogeneous dough. The overall amount of water used was 720 mL for 1 kg of dried ingredients (whole wheat flour and protein ingredients). The samples were coded as: Cs, WPs, WPBs, WPHs.

In the third baking process in one phase, the dried sourdough was added directly into the dough (2.5%). The yeast (2.5%) and the salt (1.5%) were dissolved in water and added to the mixture. The amount of water was calculated through Mixolab measurements (and adjusted if necessary) and it was around 750 mL for 1 kg of dried ingredients. A homogenous dough was obtained through kneading. The samples were coded as: Cd, WPd, WPBd, WPHd.

A control sample (100% wheat flour) was also considered in all the processes (C, Cs and Cd).

In all the three processes, the ingredients were mixed in a Diosna mixer (Germany) for 3 min at a low speed and 3–5 min at a high speed. The dough was allowed to rest for 15 min and then divided into pieces of 575–580 g (in order to have a final end product of around 500 g), rounded and left to rest for 10 min. The samples were then molded in the aluminum bread pans and proofed in a proofer at 38 °C and 44% humidity for 20–25 min (MCE Meccanica, Italy). The samples were baked in an oven (Mondial Forni, Italy) for 20–25 min at 220–230 °C. In the first 4 s, steam was used inside the oven. The breads were cooled down for 2 h at room temperature and were stored in polypropylene bags for analysis.

#### 2.2.3. Compositional Analysis of the Bread

The characteristics of the raw materials and bread samples were determined as follows: protein content by Kjeldahl method with a conversion factor of nitrogen to protein of 6.25 (AOAC 979.09) [20]; fat content by extraction with petroleum ether under reflux conditions in a Soxhlet (AOAC 963.15) [21]; ash by gravimetric method by burning at 550 °C in a furnace (AOAC 923.03) [22]. Total carbohydrate was calculated as a percentage by subtraction, as 100 − (%protein + %fat + %fiber + %ash). Calorie contents were calculated using the following conversion factors: 9 for fat, 4 for carbohydrates, 4 for protein [23].

#### 2.2.4. Bread Crumb Texture Analysis

The texture properties of the crumb were measured with the Instron Texture Analyzer (model 5944, Illinois Tool Works Inc., Norwood, MA, USA), using a 12 mm diameter compression piston and a 50 N load cell. The test was performed at room temperature, as one test in the middle of each sample (a 20 mm slice of bread), immediately after its formation. The method included a cycle of two compressions of 20 mm depth with a compression speed of 2 mm/min. Three measurements were performed for each sample. Using the Bluehill 3.13 program, the texture parameters were calculated: hardness, expressed in N (strength 40%); elasticity; cohesiveness; chewiness (or gumminess). Hardness is the force necessary to attain a given deformation and indicates the maximum force required to compress the bread to 40% of its original height. Elasticity is the rate of which a deformed material returns to the undeformed condition after the force is removed. Crumb elasticity was measured by the distance of the detected height during the second compression divided by the original compression distance (Distance 2/Distance 1). Cohesiveness is the strength of the internal bonds making up the body of the product and it was calculated as the area of work during the second compression divided by the area of work during the first compression (Area2/Area1). Chewiness was calculated as the product of hardness × cohesiveness × elasticity.

#### 2.2.5. Bread Crumb Color Analysis

The color of the bread crumb samples was measured with Konica Minolta Colorimeter (Spectrophotometer CM-5, Osaka, Japan). Three parameters were determined: parameter L*—the brightness of the sample on a scale from 0 to 100 (0 is black and 100 is white); parameter a*—the color of the sample on the scale from pure green to pure red (−a = green and +a = red), and parameter b*—the color of the sample on a scale from pure blue to pure yellow (−b = blue and +b = yellow). Before starting the analysis, the equipment was calibrated. The preparation of the samples consisted of filling the special glass cylinder of the device (diameter of 45 mm and a height of 17 mm) with crumb, without leaving air gaps. The sample thus prepared was placed on the measuring area and the glass was rotated in different positions to obtain 10 values for each color parameter. Means and standard deviation were calculated for each color parameter.

#### 2.2.6. Bread Volume

For the bread volume, the rapeseed displacement method was employed using the Fornet bread volumeter (Chopin, France) (AACC. Method 10–05.01) [24]. The volume of the products was expressed in cm^3^ per 100 g product and was calculated with the formula: V = V_1_/m × 100 (cm^3^/100 g product): V_1_—the determined volume of the sample used, in cm^3^, m—mass of the sample used, in grams.

#### 2.2.7. Bread Acidity

Bread acidity was performed according to Romanian regulation SR 91:2007 [25]. Acidity expressed in degrees was determined by titration of an aqueous extract of bread with 0.1 N NaOH solution in the presence of phenolphthalein as indicator.

#### 2.2.8. Mapping the Volatile Composition of Bread Samples Using Electronic Nose

The volatile composition of bread samples was mapped using the multisensor system α-Prometheus (electronic nose). The principle of the method consists in generating the headspace in the stapled vial with the sample, extracting a quantity from it (µL, quantity established in the method) using the syringe and injecting it into the SAS (System Array Sensor). 1 g of crumb samples were weighed in vials and then stapled. Three identical vials were prepared from each sample. The samples were heated and shaken in the apparatus oven at 35 °C for 600 s with a stirring speed of 500 rpm, the syringe being heated to 40 °C. From the volatile part generated inside the vial, 2000 μL were injected into the 18 sensor system for recording the total volatile fingerprint and comparing the fingerprints between the samples. PCA analysis using the Alpha Mos software allowed analysis of the differences between the samples. A high discrimination index (100) means that the samples have different volatile compositions.

#### 2.2.9. Microscopic Analysis of Bread Samples

The microstructure of the samples was analyzed using a field emission scanning electron microscope (FESEM) Gemini 500 from Zeiss (Jena, Germany). Bread crumb samples were dried and ground. The samples were fixed on supports with double adhesive tape made of Cu and subjected to analysis without metal coating. The microstructural images were recorded with the secondary electron detector at different magnifications to highlight morphological details.

#### 2.2.10. Sensory Analysis

Twelve trained panelists (nine females and three males), with an average age of 32 years old, analyzed the bread samples for the bread overall acceptability, using a nine-point hedonic scale (from 9 = like extremely to 1 = dislike extremely) [26]. The sensory evaluation study was approved by the Ethics Committee of the National Institute of Research & Development for Food Bioresources – IBA Bucharest.

#### 2.2.11. Statistical Analysis

A data analysis was carried out using ANOVA (one-way analysis of variance) with Tukey’s test (Minitab^®^19, Minitab Ltd., Coventry, UK). Differences among the samples were considered significant at *p* < 0.05. Values were expressed as mean ± standard deviation.

## 3. Results

### 3.1. Rheological Analysis of Flour Mixtures with Different Percentages of Protein Additions

The influence of the protein ingredients additions on the dough mixing properties is summarized in Table 2.

### 3.2. The Compositional Analysis of the Bread

The compositional analysis of the bread shown in Table 3 was performed to evaluate the changes brought about by the partial replacement of the whole wheat flour with protein ingredients.

### 3.3. Texture Analysis for Bread Samples

The results of the texture analysis for the samples obtained according to the recipe in Table 1 are presented in Figure 2.

### 3.4. Results of Bread Crumb Colour Analysis

The results of the color analysis for the samples obtained according to the recipe in Table 1 are shown in Figure 3.

The color of the samples was measured with a Konica Minolta Colorimeter (Spectrophotometer CM-5) where 3 parameters were determined respectively:
Parameter L*—measures the brightness of the sample on a scale from 0 to 100, where the value 0 represents black, and the value 100 represents white;Parameter a*—represents the color of the sample on the scale from pure green to pure red, where negative values are green, positive values are red, and 0 is neutral;Parameter b*—represents the position of the sample on a scale from pure blue to pure yellow, where negative values represent blue, positive values yellow and 0 is neutral.


### 3.5. Discrimination of Bread Samples by Electronic Nose System

The results obtained for the discrimination of samples using multivariate statistics for interpretation of the 18 sensors answers are shown in Figure 4. Each graphic presentes the distribution of the samples obtained through different bread making process and demonstrates the differences in the volatile composition of the samples.

### 3.6. Microstructure of Bread Crumb Samples

FESEM was used to investigate the bread samples microstructure. From Figure 5 it may be noticed the differences which appears when different bread making technological process are applied.

### 3.7. Sensory Analysis

The results regarding the total acceptability of the bread samples are presented in the Figure 6.

## 4. Discussion

The replacement of 2–3% of wheat flour with protein rich ingredients causes a dilution of the wheat flour components such as gluten and starch. Moreover, the presence of some specific components in the added ingredients such as fibers and fat (especially in the press cakes from hemp and sea buckthorn) influenced the dough rheological behaviour and the bread making characteristics.

The results presented in Table 2 show that the water absorption slightly increased for WP and WPH (*p* < 0.05). Proteins are one of the main components involved in water absorption, which varies with the type of protein [27,28]. Taherian et. al., 2011 [27] showed that the supplementation of wheat flour with 10% pea protein isolate (96.1% protein content on dry matter basis) caused an increase in water absorption. Hoehnel, 2019 [28] also observed a higher water absorption when a pea protein isolate (80.19% protein content) was added in a percentage of 15% to wheat flour. In addition to proteins, flours contain large amounts of starch and non-starch polysaccharides, including fiber, which significantly alter the binding of water and influence the rheological properties of the final dough [29]. In the case of sample WPB, the water absorption slightly decreased because the ingredient from sea buckthorn press cake had a protein content similar to whole wheat flour but a higher content in lipids and ash (cellulose). The development time of the dough varied between 7.59 min (WPB) and 8.49 min (WPH) but with no significant differences (the means shared the same letter in the column; *p* > 0.05). The best stability of the dough was found for control C (10.13 min) WP (9.95 min) and WPH (9.46 min.), with no significant changes (*p* > 0.05). On the other hand, sample WPB showed a significantly lower dough stability: 8.34 min (*p* < 0.05). The decrease of the stability caused by the replacement of 2% of whole wheat flour with the sea buckthorn ingredients from press cake can be explained by the fact that the sea buckthorn ingredient used in the experiments brought different proteins and fibers into the wheat dough diluting the viscoelastic network formed by gluten proteins. This behavior was found in previous studies as well when different fibers or non-traditional ingredients were used for bread making [30,31,32]. Samples WP and WPB recorded slightly lower values for C2 Mixolab parameter than the control (*p* < 0.05), which represents a lower resistance in the protein network. The gelatinization of starch (C3 Mixolab parameter) was slightly lower in sample WPB (1.85 Nm) compared with 1.90–1.94 Nm for the rest of the samples (*p* < 0.05). The stability of the starch gel (C4 Mixolab parameter) varied from 1.36 to 1.42 Nm but there were no significant differences between the samples (*p* > 0.05).

The physicochemical parameters of the bread samples are presented in Table 3. The protein content of the samples slightly increased, especially for samples with pea protein concentrate. WPs had the highest protein content (10.32%). Control C had the lowest protein content (9.34%). The protein contents of the samples with sea buckthorn press cake ingredient and hemp press cake ingredient were only slightly improved because the protein content of these ingredients was lower. The fat content did not change significantly between the samples (*p* > 0.05). The calculated caloric value of the samples had, in general, a lower value (*p* < 0.05) than the control bread. Using whole wheat flour, which is rich in proteins and fiber, as the basis of the samples, all the obtained samples had at least 12% of the energy value of the bread provided by protein, so all samples can be considered a “source of protein”. Furthermore, it was demonstrated that the addition of 2% ingredients from hemp and sea buckthorn press cake could deliver sensorially acceptable bread samples and this can represent a possibility to valorize these by-products which are valuable in bioactive compounds, and to improve the nutritional value of bread. From a sensory perspective, the whole wheat flour allowed the addition of hemp and sea buckthorn press cakes ingredients because it had higher ash content and a darker color similar with the by-products used. On the other hand, pea protein concentrate had a higher protein content, a white color and a better influence on dough rheology. As such, a combination between pea concentrate and hemp and sea buckthorn by-products was found as a solution to obtain bread enriched in protein with acceptable sensory attributes.

The bread volume increased in almost all the samples with plant protein ingredients obtained through the indirect bread making process (in two phases) and in the bread making process with dried sourdough. The use of dried sourdough ingredient had a positive effect on the volume which increased from 222 to 247 cm^3^/100 g. For samples with pea protein concentrate, both bread making technologies applied (indirect in two phases and direct with dried sourdough) had a good effect on increasing the volume. The volume increased more for WPd, from 211 to 243 cm^3^/100 g (*p* < 0.05) (Figure 1).

The acidity of the samples with sponge and sourdough was higher. The samples with sea buckthorn ingredient had a higher acidity due to the fact that sea buckthorn is a very good source of phenolic compounds such asphenolic acids, flavonoids and proanthocyanidins [3]. The samples had an astringent and acid taste (specific for sea buckthorn fruit) and a brown color.

Figure 2 presents the changes in the texture parameters (hardness, elasticity, cohesiveness and chewiness) of the bread samples as a function of the ingredients and the bread making process.

The hardness of the samples was slightly increased by the addition of vegetable protein ingredients, rising from 4.22 N for C to 5.14 N for WP (*p* < 0.05). This is similar with previous findings when the addition of 15% pea protein ingredient (with 80.19% protein content) increased the hardness of the samples [24]. The two phase bread making process (sponge and dough) had a very good effect on decreasing the hardness of the crumb and increasing the elasticity for samples C, WP, WPH. The hardness decreased more than 70% for WP and more than 50% for WPH (*p* < 0.05). In the case of WPB, the hardness did not change significantly in the samples obtained in the two phase bread making process compared with the direct bread making process (*p* > 0.05). In the samples WPBd (with sea buckthorn and dried sourdough) a different effect was observed: the crumb hardness increased from 4.42 N to 5.41 N (*p* < 0.05).

The elasticity did not vary significantly (*p* > 0.05) with the addition of plant protein ingredients. However, there was a visible improvement in the crumb elasticity in the bread making process using two phases and with dried sourdough for samples C, WP, WPH.

Cohesiveness can be explained as the strength of the internal bonds in the bread crumb which keep the crumb particles together during the mastication process. The values for cohesiveness (Figure 2) varied from 0.29 (for C), 0.26 (WP, WPH) to 0.49 (Cs), 0.57 (WPs) (*p* < 0.05). This demonstrates that cohesiveness was improved when the two phase bread making process was applied (sponge and dough) or when dried sourdough was added to the dough.

Chewiness (N) is calculated by the texture analyzer software function using hardness, elasticity and cohesiveness. In sensory evaluation, chewiness can be evaluated as the number of chews necessary to reduce the sample to a state ready for swallowing. This textural parameter was influenced by the technological process applied in bread making. Chewiness decreased (Figure 2) in the samples obtained through the indirect bread making process for samples C, WP and WPH with 30–50% (*p* < 0.05) but increased for sample WPB. The highest chewiness was recorded for the sample WPB, and the lowest values for sample Cs and WPs.

The bread making technological process influenced the color of the samples as well. Figure 3 shows that the lightness (L*) of samples slightly decreased when an indirect bread making process was applied (sponge and dough) and when dried sourdough was added to the raw ingredients. A significant change in the color was noted for the samples with sea buckthorn obtained through different bread making technologies. The use of the indirect process with a sponge phase and the addition of dried sourdough caused a significant reduction in the lightness (*p* < 0.05). Overall, the darkest samples were the samples containing sea buckthorn as an ingredient (WPB, WPBs, WPBd). There were no significant differences between the samples obtained through the direct process: C, WP, WPH, WPB, probably because the use of whole wheat flour which masked the addition of plant ingredients. The color parameter a* increased for samples WP and WPB compared with C, which means that the addition of pea protein and sea buckthorn ingredient determined an intensification of the red hue of the samples. The application of the indirect bread making process and the use of dried sourdough also led to an accentuation of the red shade. For samples with hemp ingredient, a smaller value of the color parameter a*was observed compared with samples C, WP and WPB (*p* < 0.05), which means that these samples had a greener tint. Color parameter b* had no important variations for the samples obtained through the same bread making process (*p* > 0.05). However, the applied technology determined changes in the values of b* color parameter. For samples Cs, WPs, WPBs and WPHs, the value of parameter b* was less. This means that using the indirect process or dried sourdough gave the bread crumb a blue hue.

E-nose technique was used to assess the differences between the global volatile composition of the samples and check if the samples can be differentiated in terms of the volatiles (smell) when different bread making technologies are applied. Multivariate Statistics Principal Component Analysis (PCA) was used to evaluate the differences based on the calculation of a discrimination index (AlphaMos Software). Figure 4 shows the PCA plot that represents a map of the discrimination of the bread samples. The results were compared in groups: samples C, Cs and Cd on graphic (a), with a discrimination index of 88; samples WP, WPs and WPd on graphic (b) with a discrimination index of 100; samples WPB, WPBs and WPBd on graphic (c) with a discrimination index of 98, and samples WPH, WPHs and WPHd on graphic d) with a discrimination index of 100. In all cases, the volatile composition of the samples was different as the discrimination index was very high. The samples obtained through the indirect bread making process (with sponge) are separated in the right part of the graphic and the samples prepared through the direct process and with dried sourdough are separated on the left part but in a different chart quadrant. This demonstrates that the bread making technological process has an important effect on the bread’s volatile flavor.

Scanning electron microscopy (Figure 5) was performed on bread crumb to determine the effects of protein ingredients on the gluten network. Without an added protein ingredient, the starch was well wrapped and covered by gluten (Figure 5: A, samples obtained through the direct baking process; B, samples prepared with sponge; C-samples obtained using dried sourdough). It could be seen that the samples obtained in the two phases bread making process (B) were more homogenous. With the addition of 1–2% protein ingredient from pea, sea buckthorn or hemp, the structure of the gluten network began to break and some starch particles were exposed, and small holes appeared. The addition of protein ingredients destroyed the gluten network structure. Samples obtained with dried sourdough (Cd, WPd, WPBd and WPHd) showed some dried particles, and the dough seemed to be insufficiently homogenous. The ingredients from sea buckthorn and hemp had a high fiber content which increased the roughness of the dough.

Regarding the overall sensory acceptability (Figure 6), all the samples were different (*p* < 0.05). The most preferred samples were: WPd > Cd > WPHd. Less preferred samples were WPB and WPBs, probably because of the astringent and sour taste, higher acidity and reduced volume of the bread samples containing sea buckthorn. The use of dried sourdough significantly improved the overall quality of samples with sea buckthorn. WPBd received a good score (5.83) compared with 4.67 for WPB and WPBs.

## 5. Conclusions

In present study, whole wheat bread “sources of protein” were obtained using plant proteins from pea, hemp and sea buckthorn press cakes as ingredients. The addition of plant proteins to bread had a negative impact on the hardness of the samples, which increased with their addition. Moreover, the color of the samples was changed even if the basis of the samples was whole wheat flour. Especially for samples with the sea buckthorn ingredient (WPB, WPBs, WPBd) the color was darker. With the addition of 1–2% pea, sea buckthorn or hemp protein ingredients, the gluten network structure started to break and some starch particles were exposed and small holes appeared. The sea buckthorn and hemp ingredients had a high fiber content which resulted in a rougher dough. The bread making process in two phases (with sponge and dough) improved the sensory attributes of the samples. For samples C, WP, WPH, the indirect bread making technological process with sponge and dough had a very good effect on the texture of the crumb, decreased the hardness, increased the elasticity and improved the cohesiveness of the samples. However, the samples obtained with only pea concentrate had the best sensory acceptability scores due to the fact that the pea ingredient was a white powder, with particle size similar to flour particles, and with a higher protein content which imposed the necessity of use of a small amount in the recipe. Overall, the applications presented in this paper show possibilities to valorize the hemp and sea buckthorn by-products obtained from press cakes in bread as a “source of protein” by applying the indirect bread making process or sourdough use for best results.

## Figures and Tables

**Figure 1 foods-11-03667-f001:**
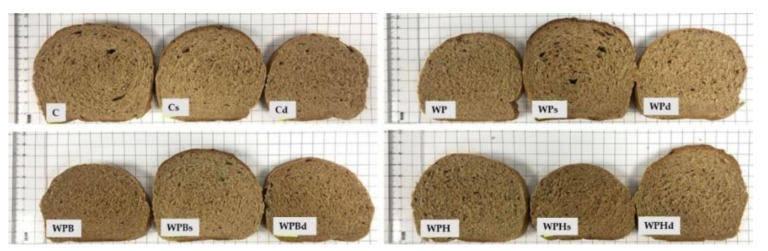
Bread samples.

**Figure 2 foods-11-03667-f002:**
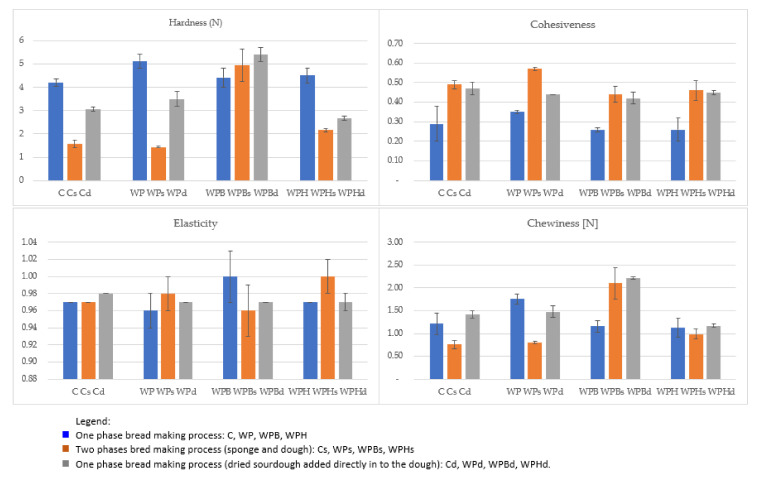
Texture properties of bread samples.

**Figure 3 foods-11-03667-f003:**
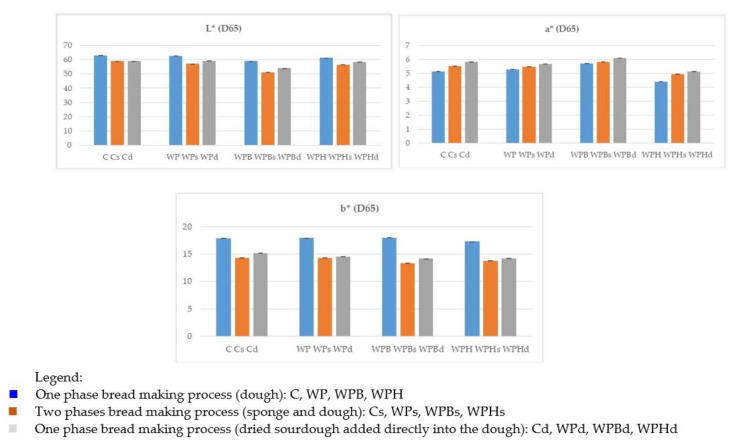
The results of the color analysis of the bread samples.

**Figure 4 foods-11-03667-f004:**
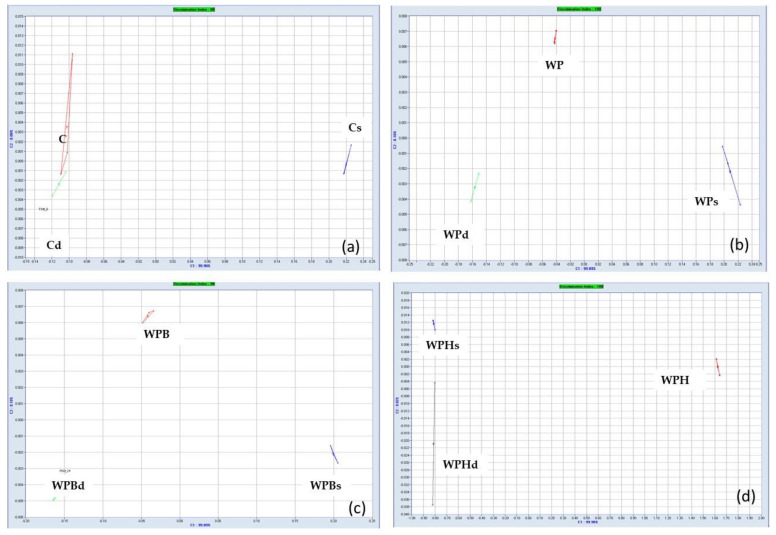
Principal component analysis (PCA) plot for the bread samples. (**a**) Discrimination of samples C, Cs, Cd; (**b**) Discrimination of samples WP, WPs, WPd; (**c**) Discrimination of samples WPB, WPBs, WPBd; (**d**) Discrimination of samples WPH, WPHs, WPHd.

**Figure 5 foods-11-03667-f005:**
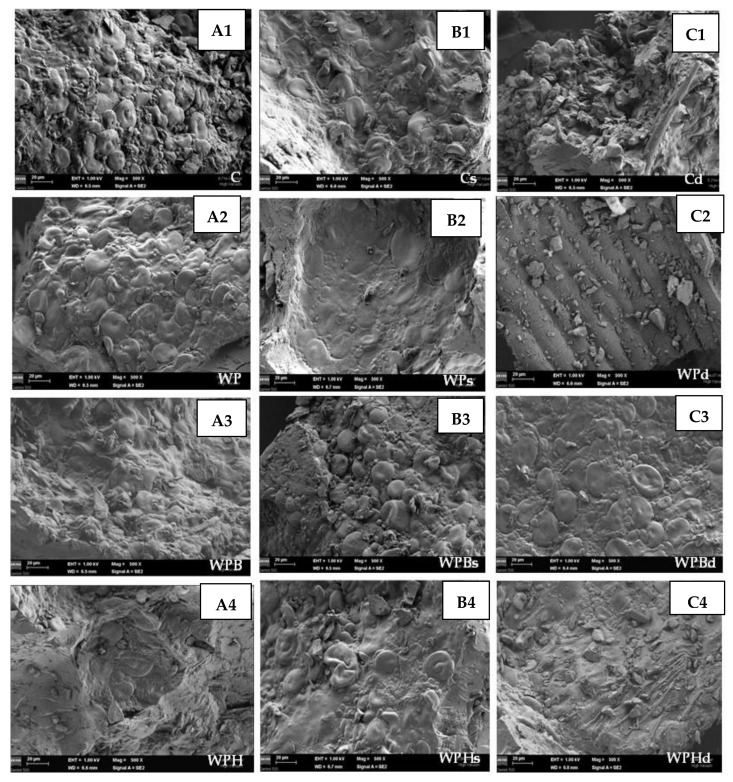
Microstructure of bread crumb samples. (**A1**–**A4**) samples obtained through direct bread making process; (**B1**–**B4**) samples obtained through indirect bread making process (sponge and dough); (**C1**–**C4**) samples obtained through indirect bread making process with dried sourdough. C-control from whole wheat flour; WP-whole wheat flour + 2% pea concentrate; WPB-whole wheat flour + 1% pea concentrate + 2% sea buckthorn ingredient; WPH-whole wheat flour + 1% pea concentrate + 2% hemp ingredient.

**Figure 6 foods-11-03667-f006:**
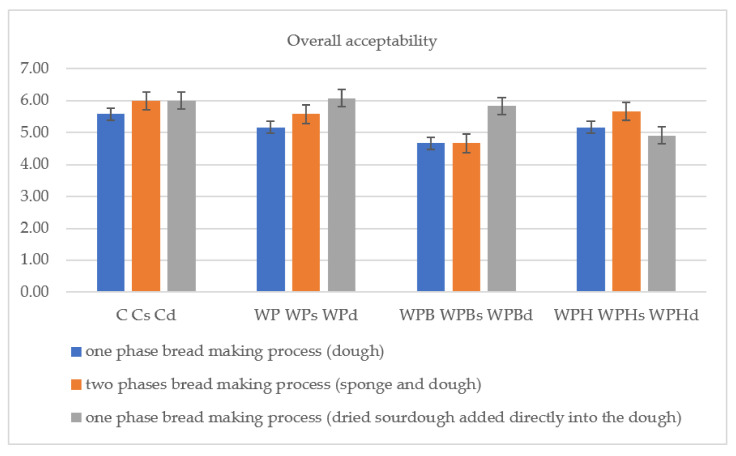
Overall acceptability.

**Table 1 foods-11-03667-t001:** Bread formulation.

Samples	Whole Wheat Flour (g)	Pea Protein Concentrate (g)	Hemp Press Cake (g)	Sea Buckthorn Press Cake(g)	Yeast (g)	Dried Sourdough (g)	Salt (g)	Water (mL)
Bread samples made by direct method								
C-control from whole wheat flour	1000	-	-	-	30	-	15	660
WP-whole wheat flour + 2% pea concentrate	980	20	-	-	30	-	15	690
WPB-whole wheat flour + 1% pea concentrate + 2% sea buckthorn ingredient	970	10	20	-	30	-	15	660
WPH-whole wheat flour + 1% pea concentrate + 2% hemp ingredient	970	10	-	20	30	-	15	670
Bread samples made by indirect method (sponge and dough)								
Cs-control from whole wheat flour	1000	-	-	-	20	-	15	720
WPs-whole wheat flour + 2% pea concentrate	980	20	-	-	20	-	15	700
WPBs-whole wheat flour + 1% pea concentrate + 2% sea buckthorn ingredient	970	10	20	-	20	-	15	720
WPHs-whole wheat flour + 1% pea concentrate + 2% hemp ingredient	970	10	-	20	20	-	15	700
Bread samples made by direct method (with dried sourdough)								
Cd-control from whole wheat flour	1000	-	-	-	25	25	15	750
WPd-whole wheat flour + 2% pea concentrate	980	20	-	-	25	25	15	750
WPBd-whole wheat flour + 1% pea concentrate + 2% sea buckthorn ingredient	970	10	20	-	25	25	15	750
WPHd-whole wheat flour + 1% pea concentrate + 2% hemp ingredient	970	10	-	20	25	25	15	750

The sign “-” represents the non-use of that ingredient in the recipe, it can be considered to be the value "0".

**Table 2 foods-11-03667-t002:** Rheological analysis of flour mixtures with different percentages of protein additions.

Samples *	Moisture (%)	Water Absorption (%)	Development Time (min)	Stability (min)	C2 (Nm)	C3 (Nm)	C4 (Nm)	C5 (Nm)
C	12.21 ± 0 ^c^	65.80 ± 0 ^c^	8.06 ± 0.13 ^a^	10.13 ± 0.21 ^a^	0.40 ± 0 ^b^	1.94 ± 0 ^a^	1.4 ± 0.03 ^a^	2.08 ± 0.1 ^a^
WP	12.40 ± 0 ^a^	68.60 ± 0 ^a^	8.17 ± 0.52 ^a^	9.95 ± 0.21 ^a^	0.38 ± 0 ^c^	1.90 ± 0 ^a,b^	1.36 ± 0.01 ^a^	1.92 ± 0.05 ^a^
WPB	12.30 ± 0 ^b^	65.40 ± 0 ^d^	7.59 ± 1.53 ^a^	8.34 ± 0.02 ^b^	0.33 ± 0 ^d^	1.85 ± 0 ^b^	1.39 ± 0.05 ^a^	2.25 ± 0 ^a^
WPH	12.30 ± 0 ^b^	66.50 ± 0 ^b^	8.49 ± 0.58 ^a^	9.46 ± 0.13 ^a^	0.40 ± 0 ^a^	1.94 ± 0.02 ^b^	1.42 ± 0 ^a^	1.55 ± 0.49 ^a^

* Samples: C-control from whole wheat flour; WP-whole wheat flour + 2% pea concentrate; WPB-whole wheat flour + 1% pea concentrate + 2% sea buckthorn ingredient; WPH-whole wheat flour + 1% pea concentrate + 2% hemp ingredient. Values are expressed as mean ± standard deviation (*n* = 2). Means that do not share a letter in a column are significantly different (*p* < 0.05).

**Table 3 foods-11-03667-t003:** Compositional analysis of breads (expressed in % of the fresh bread).

Samples	Moisture (%)	Protein (%)	Fat (%)	Ash (%)	Carbo Hydrates * (%)	Energy (kcal/100 g)	Energy from Protein ** (%)	Volume (cm^3^/100 g)	Acidity (°)
C	45.08 ± 0.05 ^f,g^	9.34 ± 0.08 ^e,f^	0.11 ± 0.01 ^a^	1.45 ± 0.01 ^a,b,c^	44.02 ± 0.11 ^a,b^	214.43 ± 0.13 ^c^	17.41 ± 0.09 ^g^	222 ± 1 ^d^	1.8
Cs	44.49 ± 0.05 ^h^	9.56 ± 0.05 ^d,e^	0.11 ± 0.02 ^a^	1.35 ± 0.05 ^a,b,c^	44.48 ± 0.05 ^a^	217.18 ± 0.10 ^a^	17.61 ± 0.10 ^f,g^	226 ± 2 ^d^	1.8
Cd	45.33 ± 0.06 ^e,f^	9.21 ± 0.05 ^f^	0.12 ± 0.05 ^a^	1.45 ± 0.05 ^a,b,c^	43.89 ± 0.16 ^a,b,c^	213.46 ± 0.05 ^d^	17.26 ± 0.15 ^g^	247 ± 1 ^b^	2.0
WP	45.65 ± 0.07 ^c,d^	9.99 ± 0.09 ^a,b^	0.10 ± 0.05 ^a^	1.30 ± 0.05 ^b,c^	42.96 ± 0.12 ^e,f,g^	212.68 ± 0.10 ^e^	18.79 ± 0.05 ^a,b,c^	211 ± 1 ^e^	1.2
WPs	45.57 ± 0.05 ^c,d,e^	10.32 ± 0.05 ^a^	0.31 ± 0.05 ^a^	1.47 ± 0.05 ^a,b,c^	42.34 ± 0.11 ^g^	213.40 ± 0.12 ^d^	19.34 ± 0.13 ^a^	224 ± 2 ^d^	2.0
WPd	47.63 ± 0.05 ^a^	9.56 ± 0.05 ^d,e^	0.33 ± 0.05 ^a^	1.28 ± 0.05 ^c^	41.20 ± 0.10 ^h^	206.02 ± 0.15 ^h^	18.57 ± 0.05 ^b,c,d^	243 ± 2 ^b^	2.0
WPB	45.86 ± 0.06 ^c^	9.61 ± 0.09 ^b,c,d^	0.22 ± 0.05 ^a^	1.52 ± 0.05 ^a,b^	42.78 ± 0.15 ^f,g^	211.58 ± 0.10 ^f^	18.17 ± 0.12 ^d,e,f^	194 ± 3 ^f^	2.8
WPBs	45.51 ± 0.07 ^d,e^	9.70 ± 0.05 ^b,c,d^	0.16 ± 0.05 ^a^	1.48 ± 0.01 ^a,b,c^	43.15 ± 0.10 ^d,e,f^	212.84 ± 0.10 ^e^	18.23 ± 0.09 ^c,d,e^	204 ± 2 ^e,f^	3.4
WPBd	45.50 ± 0.05 ^d,e^	9.66 ± 0.05 ^b,c,d,e^	0.17 ± 0.05 ^a^	1.42 ± 0.02 ^a,b,c^	43.25 ± 0.10 ^c,d,e,f^	213.19 ± 0.14 ^h^	18.12 ± 0.08 ^d,e,f^	239 ± 1 ^b,c^	2.2
WPH	47.30 ± 0.05 ^b^	9.93 ± 0.05 ^b,c^	0.20 ± 0.05 ^a^	1.51 ± 0.05 ^a,b,c^	43.45 ± 0.12 ^b,c,d,e^	215.28 ± 0.13 ^b^	18.45 ± 0.12 ^b,c,d^	231 ± 2 ^c,d^	1.4
WPHs	44.86 ± 0.06 ^g^	9.74 ± 0.05 ^b,c,d^	0.16 ± 0.05 ^a^	1.57 ± 0.06 ^a^	41.64 ± 0.13 ^h^	206.99 ± 0.19 ^g^	18.83 ± 0.13 ^c,d,e^	286 ± 1 ^a^	2.2
WPHd	47.03 ± 0.05 ^b^	9.54 ± 0.05 ^b,c,d,e^	0.20 ± 0.05 ^a^	1.36 ± 0.05 ^a,b,c^	43.67 ± 0.11 ^b,c,d,e^	214.63 ± 0.09 ^c^	17.79 ± 0.07 ^d,e,f^	199 ± 2 ^f^	1.8

* Calculated by difference. ** Indicates the energy provided by the proteins from the whole energy content. All the results are mean ± standard deviation (*n* = 3). Values followed by different superscript letters in the same column are significantly different (*p* < 0.05). Means that do not share a letter in a column are significantly different (*p* < 0.05). Samples: C-control from whole wheat flour through direct bread making process, Cs-control from whole wheat flour with sponge (two phase bread making process), Cd-control from whole wheat flour with dried sourdough; WP-whole wheat flour + 2% pea concentrate through direct bread making process, WPs-whole wheat flour + 2% pea concentrate with sponge (two phases bread making process); WPd-whole wheat flour + 2% pea concentrate with dried sourdough; WPB-whole wheat flour + 1% pea concentrate + 2% sea buckthorn ingredient through direct bread making process; WPBs-whole wheat flour + 1% pea concentrate + 2% sea buckthorn ingredient with sponge; WPBd-whole wheat flour + 1% pea concentrate + 2% sea buckthorn ingredient with dried sourdough, WPH-whole wheat flour + 1% pea concentrate + 2% hemp ingredient through direct bread making process, WPHs-whole wheat flour + 1% pea concentrate + 2% hemp ingredient with sponge, WPHd-whole wheat flour + 1% pea concentrate + 2% hemp ingredient with dried sourdough.

## Data Availability

Data is contained within the article.

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
