# Peer review of "The Influence of the Technological Process on Improving the Acceptability of Bread Enriched with Pea Protein, Hemp and Sea Buckthorn Press Cake"

_foods, 2022, doi:10.3390/foods11223667_

Round 1

Reviewer 1 Report

Comments for Authors

I reviewed the current research paper entitled “The influence of the technological baking process on improving the acceptability of plant enriched baked products”. The subject is interesting and the presented physico-chemical, rheological and sensorial characteristics of various plant protein products compared to wheat flour is a new issue. The concept is interesting and useful and the authors have put an effort in providing explanations and have commented in previous findings of the international literature.  In my opinion, the manuscript is inaccurately prepared and requires many corrections and supplements. Some are;

Title could be simplified, for example: “The influence of the technological baking process on improving the acceptability of plant enriched bread”

Introduction: This section must be reorganized. Some parts must be reduced so more focus can be given to the protein plant sources and enrichment in bread.

Please mention previous studies regarding plant protein and their benefits or its contribution to bakery products.

Rewrite the methods “ 2.2.3. Compositional analysis of the bread samples”. The AACC methods number should be added and each method could be briefly addressed.

Table-1. authors should include in the description of the tables, the statistical analysis performed and lettering mean, etc. 6. In table1, footnote was missing.

Tables 2: please specify as a footnote which are the meaning of the superscript letter

Add the Unit in Cohesiveness and Elasticity (Figure 3.3 Texture analysis of bread)

How the dough stability time and Dough development time decreased? Please explain.

Conclusions must be more focused and targeted on the overall outcome of the study. The present section is a summary of the results.

The quality of written English is not bad but can be improved.

The current paper is highly Plagiarized >32%. Plagiarism should be removed according to journal instructions.  

Reviewer 2 Report

This manuscript was difficult to follow although the authors studied on an important subject. English language could have been controlled by the professions of that language. 

*The present study is related to protein enriched bread not baked products. Baked/bakery products are common name of the products including bread, cake, biscuits etc. Also, baking process refers the process of the baking/cooking in the oven. Different time and temperature conditions and baking type can be studied in this context. In this study, this is not baking process, this is bread dough making process. These are considered as a mistake which shows that the authors didn’t understand the differences between bakery products and their process. 

* ‘’The 21 energy from protein varied from 17.26 to 19.34 % which means that all the samples can be considered a source of protein”’’.  This is an expected result because protein sources are incorporated into formulation. Instead of this, authors should have been explained that the effects of increasing protein content on bread quality, acceptability beside rheological properties of the dough. 

*The authors had given less information about why they chose these proteins and what differences are exist when compared to other source of protein. 

*  Why whole wheat flour was used in this study rather than other types of flours? Whole wheat bread is already a healthy bread type. It contains vitamins, mineral, fibers.. etc. Especially, sourdough whole wheat bread contains bioactive compounds such as antioxidants and proteins. Why whole wheat bread was chosen in order to enrich pea, hemp and sea buckthorn proteins?  It should have been explained. 

*  Why did all formulation contain pea protein? What was the reason of it?

*  How were the ratios (2% pea protein concentrate; 1% pea + 2% sea buck- 15 thorn ingredients; 1% pea + 2% hemp ingredients) determined and chosen?

 *  Why sourdough bread making process were preferred? And, what was the purpose of using dried sourdough? All them should have been clarified.

*  In indirect process in two phases method, sourdough was prepared with using half of the dried ingredients …… This is not a sourdough, this is a sponge dough process. In sourdough bread making process, flour and water are mixed and fermented app. 16-24 h. And There are different kinds of sourdough making methods and different types of sourdough (Type I, II, III and IV etc.).

*  The results section contains only table and figures. It should be expanded and most prominent and statistically significant results should have been explained according to the tables and figures. Also, p>0.05 and p<0.05 should have been included in appropriate places.

*  In table 2, volume or specific volume? It should have been calculated as specific volume (cm^3/g). 

Round 2

Reviewer 1 Report

The authors improved the manuscript according to the comments of the reviewer.

Reviewer 2 Report

Although the authors revised the manuscript, the bread making methods are still not clear and difficult to understand. The results section wasn't revised sufficiently. However, it is still difficult to follow. 
